# Asymptotics of $\ell_2$ Regularized Network Embeddings

**Andrew Davison**
Department of Statistics
Columbia University
New York, NY 10027
`ad3395@columbia.edu`

## Abstract

A common approach to solving prediction tasks on large networks, such as node classification or link prediction, begins by learning a Euclidean embedding of the nodes of the network, from which traditional machine learning methods can then be applied. This includes methods such as DeepWalk and node2vec, which learn embeddings by optimizing stochastic losses formed over subsamples of the graph at each iteration of stochastic gradient descent. In this paper, we study the effects of adding an $\ell_2$ penalty of the embedding vectors to the training loss of these types of methods. We prove that, under some exchangeability assumptions on the graph, this asymptotically leads to learning a graphon with a nuclear-norm-type penalty, and give guarantees for the asymptotic distribution of the learned embedding vectors. In particular, the exact form of the penalty depends on the choice of subsampling method used as part of stochastic gradient descent. We also illustrate empirically that concatenating node covariates to $\ell_2$ regularized node2vec embeddings leads to comparable, when not superior, performance to methods which incorporate node covariates and the network structure in a non-linear manner.

## 1 Introduction

Network embedding methods [see e.g 6, 24–27, 48, 58, 64] aim to find a latent representation of the nodes of the network within Euclidean space, in order to facilitate the solution of tasks such as node classification and link prediction, by using the produced embeddings as features for machine learning algorithms designed for Euclidean data. For example, they can be used for recommender systems in social networks, or to predict whether two proteins should be linked in a protein-protein interaction graph. Generally, such methods obtain state-of-the-art performance on these type of tasks.

A classical approach to representation learning is to use the principal components of the Laplacian of the network [6]; however, this approach is computationally prohibitive for large datasets. In order to scale to large datasets, methods such as DeepWalk [48] and node2vec [25] learn embeddings via optimizing a loss formed over stochastic subsamples of the graph. Letting $\mathcal{G} = (\mathcal{V}, \mathcal{E})$ denote an undirected graph, and writing $\omega_u \in \mathbb{R}^d$ for the embedding of a vertex $u \in \mathcal{V}$ with embedding dimension $d$ such that $d \ll |\mathcal{V}|$, these methods learn embeddings by iterating over the following process: we take a random walk $(u_i)_{i \leq k}$ over the graph, form a loss

$$\mathcal{L} = -\sum_{i=1}^{k} \sum_{j \,:\, |j-i| \leq W} \log(\sigma\langle \omega_{u_i}, \omega_{u_j} \rangle) - \sum_{i=1}^{k} \sum_{l=1}^{L} \mathbb{E}_{v_l \sim \mathcal{N}(\cdot | u_i)} \Big[ \log(1 - \sigma(\langle \omega_{u_i}, \omega_{v_l} \rangle)) \Big], \quad (1)$$

and then perform gradient updates of the form $\omega_u \leftarrow \omega_u - \eta \nabla_{\omega_u} \mathcal{L}$ for some step size $\eta > 0$. Here $\langle \cdot, \cdot \rangle$ denotes the Euclidean inner product, $W$ is a window size, $\mathcal{N}(\cdot | v)$ a negative sampling distribution for vertex $v$ and $\sigma(y) := (1 + e^{-y})^{-1}$ the sigmoid function.

36th Conference on Neural Information Processing Systems (NeurIPS 2022).

Other methods take variations on this approach - for example, LINE [58] replaces the random walk by sampling edges uniformly from the edge set of the graph. One can generalize (1) to capture both these cases, by considering embeddings learned via stochastic updates

$$\omega_u \leftarrow \omega_u - \eta \nabla_{\omega_u} \mathcal{L} \quad \text{where} \quad \mathcal{L} = \sum_{(i,j) \in \mathcal{P}} \ell_{\mathcal{P}}(\langle \omega_i, \omega_j \rangle) + \sum_{(i,j) \in \mathcal{N}} \ell_{\mathcal{N}}(\langle \omega_i, \omega_j \rangle), \qquad (2)$$

$\mathcal{P}, \mathcal{N} \subseteq \mathcal{V} \times \mathcal{V}$ are random subsamples of the graph called positive and negative samples respectively [see e.g 32], and $\ell_{\mathcal{P}}(\cdot)$ and $\ell_{\mathcal{N}}(\cdot)$ are chosen to force the embeddings of vertex pairs appearing in $\mathcal{P}$ close together, and those in $\mathcal{N}$ far away. Usually one takes $\ell_{\mathcal{P}}(y) = -\log \sigma(y)$ and $\ell_{\mathcal{N}}(y) = -\log \sigma(-y)$. For example, in node2vec $\mathcal{P}$ is formed by taking concurrent edges in a random walk, and in LINE, $\mathcal{P}$ is formed by uniform edge sampling; in both methods $\mathcal{N}$ is taken to be the same negative sampling distribution $\mathcal{N}(u|v) \propto \deg(v)^{3/4}$ [46]. The scheme in (2) attempts to minimize

$$\mathcal{R} := \sum_{i,j} \left\{ \mathbb{P}\big((i,j) \in \mathcal{P}\big) \ell_{\mathcal{P}}(\langle \omega_i, \omega_j \rangle) + \mathbb{P}\big((i,j) \in \mathcal{N}\big) \ell_{\mathcal{N}}(\langle \omega_i, \omega_j \rangle) \right\} \qquad (3)$$

obtained by averaging over the random process used to create $\mathcal{P}$ and $\mathcal{N}$ at each iteration of stochastic gradient descent [55, 63]; see Appendix B.1 for a detailed derivation.

In the case where $\ell_{\mathcal{P}}(y) = -\log \sigma(y)$ and $\ell_{\mathcal{N}}(y) = -\log \sigma(-y)$, with $\mathcal{P}$ and $\mathcal{N}$ being random subsets of $\mathcal{E}$ and $(\mathcal{V} \times \mathcal{V}) \setminus \mathcal{E}$ (such as in LINE, or node2vec with a window length of 1), then we can view (3) as the function obtained via trying to minimize the negative log-likelihood (equivalent to maximizing the log-likelihood) of a probabilistic model

$$a_{uv} \,|\, \omega_u, \omega_v \sim \text{Bernoulli}(\sigma(\langle \omega_u, \omega_v \rangle)) \text{ independently for } u < v, \qquad (4)$$

($a_{uv} = 1$ indicating $(u, v)$ is an edge) using the stochastic gradient descent scheme specified in (2) (see Appendix B.2). If it is assumed that the embedding vectors are drawn i.i.d from some latent distribution, then this corresponds to implicitly fitting an exchangeable model to the graph [44] - see Appendix A for a brief discussion on these models. Here the distribution of the adjacency matrix $(a_{uv})_{u,v}$ is invariant to joint permutations of the vertices of the network, or equivalently (by the Aldous-Hoover theorem [1]) arise from a probabilistic model

$$\lambda_u \sim \text{Unif}([0,1]) \text{ independently}, \qquad a_{uv} \,|\, \lambda_u, \lambda_v \sim \text{Bernoulli}(W(\lambda_u, \lambda_v)) \text{ independently}, \quad (5)$$

for some symmetric measurable function $W : [0,1]^2 \to [0,1]$ called a *graphon*. We highlight that the model in (4) is implicitly fitting a dense graph model to the data, *even when the observed graph data are sparse*, as is the case with real world networks. One way of partially addressing this issue is to consider a *sparsified exchangeable graph* with $W \to \rho_n W$ for some sparsifying sequence $\rho_n \to 0$.

A natural example of a prior distribution on embedding vectors is $\omega_i \sim \text{Normal}(0, (2\xi)^{-1/2} I_d)$ for some constant $\xi > 0$, so that the contribution to the negative log-likelihood of each embedding is $\xi \|\omega_i\|_2^2$. The full negative log-likelihood of (4) with such a prior distribution is then given by

$$-\sum_{i,j} \left\{ a_{ij} \log \sigma(\langle \omega_i, \omega_j \rangle) + (1 - a_{ij}) \log \sigma(-\langle \omega_i, \omega_j \rangle) \right\} + \xi \sum_i \|\omega_i\|_2^2, \qquad (6)$$

which depends only on the matrix $G_{ij} = \langle \omega_i, \omega_j \rangle$; this loss can also arise by considering a weight decay optimization scheme [see e.g 38]. In either case, letting $\Omega \in \mathbb{R}^{n \times d}$ be the matrix whose rows are the $\omega_i$, so $G = \Omega \Omega^T$, we get that

$$\sum_{i=1}^n \|\omega_i\|_2^2 = \sum_{i=1}^n \sum_{j=1}^d \omega_{ij}^2 = \sum_{i=1}^n \sum_{j=1}^d \omega_{ij} \omega_{ji} = \sum_{i=1}^n (\Omega \Omega^T)_{ii} = \text{tr}(\Omega \Omega^T) = \text{tr}(G) = \|G\|_* \qquad (7)$$

where $\|G\|_*$ is the nuclear norm, which equals the sum of the singular values of $G$.

As a result, we can view (6) as a regularized matrix factorization problem, where the nuclear-norm penalty on the matrix $G$ is well known to shrink the singular values of $G$ exactly towards zero, and to make $G$ low rank [see e.g 4, 19, 37, 54]. This consequently lowers the effective dimension of the embeddings. From a computation perspective, this is advantageous compared to treating the embedding dimension as a tunable hyperparameter, as warm-start procedures can be used to efficiently tune the regularization weight $\xi$ (and consequently the effective embedding dimension). Tuning

the dimension optimality is also desirable, as generally networks have exact lower-dimensional factorizations than the embedding dimensions usually chosen in embedding models [16].

In this paper, our interest is in studying the effects of such a regularizer in the scenario where embeddings are learned via subsampling, in which case the corresponding version of (3) becomes

$$\sum_{i,j} \left\{ \mathbb{P}\big((i,j) \in \mathcal{P}\big)\ell_{\mathcal{P}}(\langle\omega_i,\omega_j\rangle) + \mathbb{P}\big((i,j) \in \mathcal{N}\big)\ell_{\mathcal{N}}(\langle\omega_i,\omega_j\rangle) \right\} + \sum_i \mathbb{P}\big(i \in \mathcal{V}(\mathcal{P}\cup\mathcal{N})\big)\|\omega_i\|_2^2, \quad (8)$$

where $\mathcal{V}(\mathcal{P} \cup \mathcal{N})$ is the set of vertices which appear either in $\mathcal{P}$ or $\mathcal{N}$ (see Appendix B.3). The first part of the likelihood can still be thought of as corresponding to a matrix factorization term - see e.g [50]. However, we note that for certain sampling schemes (e.g random walk samplers), the probability a vertex is sampled is not equiprobable across vertices, and so the regularizer will not be the same form as in (7). Despite this, we still want to analyze the extent to which nuclear norm type regularization (and hence effective dimension reduction) may still arise. To do so, we study minimizers $(\widehat{\omega}_1, \ldots, \widehat{\omega}_n)$ of (8) assuming the graph arises from a sparsified exchangeable graph, and obtain guarantees of the form

$$\frac{1}{n^2}\sum_{i,j}\big(\langle\widehat{\omega}_i,\widehat{\omega}_j\rangle - K(\lambda_i,\lambda_j)\big)^2 = o_p(1) \ \text{ and } \ \min_{Q\in O(d)}\frac{1}{n}\sum_i\big\|\widehat{\omega}_i - \psi(\lambda_i)Q\big\|_2^2 = o_p(1) \quad (9)$$

for some functions $K : [0,1]^2 \to \mathbb{R}$ and $\psi : [0,1] \to \mathbb{R}^d$ obtained through the process of minimizing the population objective (Section 3). Our results allow us to recover the motivation above that the regularization acts to reduce the effective dimension of the learned embedding vectors. We also illustrate experimentally that using such regularization can give performance competitive to architectures such as GraphSAGE (Section 4). We note that our theoretical results apply *in the regime* $d \ll n$, reflecting chosen embedding dimensions in practice. This means that (8) is non-convex in the matrix $G_{ij} = \langle\omega_i,\omega_j\rangle$ due to rank constraints, which complicates the theoretical analysis (as compared to e.g [50].)

## 1.1 Related works

**Guarantees for embedding methods:** We highlight that there is an extensive literature on the embeddings formed by the eigenvectors of the adjacency or Laplacian matrices of a network [e.g 3, 40–42, 56, 59]. Under various latent space models for the network, these works give guarantees on quantities of the form $\max_i \|\widehat{\omega}_i - Q\psi_i\|_2^2$ for some orthogonal matrix $Q$ and vectors $\psi_i$, to discuss recovery of latent variables and/or obtain exact recovery in a community detection task. Stronger bounds are obtained in this setting as they are able to directly apply matrix eigenvector perturbation methods to study the embeddings, which we cannot with our approach; as a tradeoff, our approach allows us to study embeddings learned via a variety of subsampling schemes, which these works do not. We highlight that the second bound in (9) can still be used to give guarantees for weak recovery of community detection [41]. There are a few works discussing random walk methods like node2vec, albeit circumventing the non-convexity in the problem; [50] discusses the unconstrained minima of the loss (1) when $d = n$, with [68] then examining the best rank $r$ approximation to this matrix when the generating model is a stochastic block model with $k$ communities and $r \le k$. [21] covers the non-convex regime $d \ll n$ in the case where $\xi_n = 0$, i.e without regularization.

**Nuclear norm penalties and $\ell_2$ regularized embeddings:** In the context of matrix factorization (so the matrix factors are embedding matrices), the effects of Frobenius norm penalties inducing nuclear norm penalties are well known [e.g 53, 61]; generally, there is an extensive literature on the effects of nuclear norm penalization in the finite-dimensional setting [e.g 4, 19, 37, 54]. In [67], it is also shown that $\ell_2$ regularized node2vec gives an improvement in performance on downstream tasks.

**Graphon estimation:** We mention that our guarantees on the gram matrix formed by the learned embeddings are similar to those obtained in the graphon estimation literature [see e.g 8, 9, 18, 22, 35, 39, 65, 66]. Depending on the choice of sampling scheme and loss, it is possible for the limiting matrix $K(\lambda_i, \lambda_j)$ to be an invertible transformation of $W(\lambda_i, \lambda_j)$, and so we compare our rates of convergence in such a scenario (see Remark 6 in Appendix D).

## 2 Framework and assumptions of analysis

Given a sequence of graphs $\mathcal{G}_n = (\mathcal{V}_n, \mathcal{E}_n)$, and writing $\boldsymbol{\omega}_n = (\omega_1, \ldots, \omega_n)$ with $\omega_u \in \mathbb{R}^d$ denoting the $d$-dimensional embedding of vertex $u$, we study the regularized empirical risk function

$$\mathcal{R}_n(\boldsymbol{\omega}_n) + \xi_n \mathcal{R}_n^{\text{reg}}(\boldsymbol{\omega}_n) \text{ where } \mathcal{R}_n(\boldsymbol{\omega}_n) := \sum_{\substack{i,j \in [n] \\ i \neq j}} \mathbb{P}\big((i,j) \in S(\mathcal{G}_n) \,|\, \mathcal{G}_n\big) \ell(\langle \omega_i, \omega_j \rangle, a_{ij}),$$

$$\mathcal{R}_n^{\text{reg}}(\boldsymbol{\omega}_n) := \sum_{i \in [n]} \mathbb{P}\big(i \in \mathcal{V}(S(\mathcal{G}_n)) \,|\, \mathcal{G}_n\big) \|\omega_i\|_2^2, \tag{10}$$

for $\xi_n \geq 0$. Here, we define a subsample $S(\mathcal{G})$ of a graph $\mathcal{G}$ as a collection of vertices $\mathcal{V}(S(\mathcal{G}))$ and a symmetric subset of $\mathcal{V}(S(\mathcal{G})) \times \mathcal{V}(S(\mathcal{G}))$. The sampling probabilities are conditional on $\mathcal{G}_n$ as we will soon assume that the $\mathcal{G}_n$ also arise from a probabilistic model. We note (10) arises from (8) whenever $\mathcal{P}$ and $\mathcal{N}$ are random subsets of $\mathcal{E}_n$ and $\mathcal{V} \times \mathcal{V} \setminus \mathcal{E}_n$ respectively (see Appendix B.4).

Throughout, $\ell(y, x)$ is either the cross-entropy loss $\ell_\sigma(y, x) := -x \log \sigma(y) - (1 - x) \log \sigma(-y)$ or the squared loss $\ell_2(y, x) := (y - x)^2$. We now discuss our assumptions on the generative model of the graph. Recall in the introduction we argued embedding methods are implicitly fitting an exchangeable graph model; consequently, as a first step to analysis, we will assume that the graph arises from a sparsified graphon (to account for the sparsity in graphs observed in the real world).

**Assumption 1.** *We assume that the sequence of graphs $(\mathcal{G}_n)_{n \geq 1}$ have vertex sets $\mathcal{V}_n = [n]$ and arise from a graphon process with generating graphon $W_n = \rho_n W$ with $\rho_n \gg \log(n)/n$, so that*

$$\lambda_i \overset{i.i.d}{\sim} \text{Unif}[0,1] \text{ for } i \in [n], \quad a_{ij} | \lambda_i, \lambda_j \overset{indep.}{\sim} \text{Bernoulli}(\rho_n W(\lambda_i, \lambda_j)) \text{ for } i < j.$$

*We moreover suppose that $W \in [c, 1-c]$ for some $c > 0$, and either a) $W$ is piecewise constant on a partition $\mathcal{Q} \times \mathcal{Q}$ where $\mathcal{Q}$ is a partition of $[0,1]$ of size $\kappa$ (so the model is a stochastic block model), or b) $W$ is Hölder$([0,1]^2, \beta_W, L_W)$ for some exponents and constants $\beta_W \in (0,1]$ and $L_W < \infty$.*

We provide an introduction to graphon models in Appendix A.

**Remark 1.** *Here we use the canonical choice of uniform latent variables for the graphon as guaranteed by the Aldous-Hoover theorem for vertex exchangeable graphs [1]; in principle, our results can extend to graphons on higher dimensional latent spaces by the same style of arguments [see e.g 21]. We highlight that our assumptions are somewhat restrictive with regards to the boundedness and and sparsity conditions; while it is common to allow $\rho_n \gg \log(n)/n$ [e.g 47, 65] - in which case the degree structure is regular, not necessarily realistic of real world networks - it is also common to work in the regime where $\rho_n = \Theta(\log(n)/n)$ or smaller [e.g 9, 66]. We highlight that in general graphons, regardless of the sparsity factor, tend to not give rise to graphs with power-law or heavy-tailed type degree structures, which is frequent with real world networks [15, 69].*

For subsampling schemes used in practice (such as random walk and uniform edge samplers), the sampling probability of vertices and edges depends only on local features of the graph. Following [21], we can formalize this as follows:

**Assumption 2.** *There exist sequences of measurable functions $(f_n)_{n \geq 1}$ and $(\tilde{g}_n)_{n \geq 1}$ and a sequence $s_n = o(1)$, such that*

$$\max_{\substack{i,j \in [n] \\ i \neq j}} \left| \frac{n^2 \mathbb{P}\big((i,j) \in S(\mathcal{G}_n) \,|\, \mathcal{G}_n\big)}{f_n(\lambda_i, \lambda_j, a_{ij})} - 1 \right| = O_p(s_n), \ \max_{i \in [n]} \left| \frac{n \mathbb{P}\big(i \in \mathcal{V}(S(\mathcal{G}_n)) \,|\, \mathcal{G}_n\big)}{\tilde{g}_n(\lambda_i)} - 1 \right| = O_p(s_n),$$

*and moreover $\mathbb{E}[f_n(\lambda_1, \lambda_2, a_{12})] = O(1)$, $\mathbb{E}[f_n(\lambda_1, \lambda_2, a_{12})^2] = O(\rho_n^{-1})$, $\mathbb{E}[\tilde{g}_n(\lambda_1)] = O_p(1)$.*

This assumption allows us to replace the sampling probabilities in the empirical risk by functions which depend on the latent variables and edge assignments in the model, from which the exchangeability in the model can be used to allow for a large sample analysis. Examples of sampling schemes satisfying this condition are given in Section 3.2. We additionally impose some regularity conditions on the "averaged" versions of the above functions defined by

$$\tilde{f}_n(l, l', 1) := f_n(l, l', 1) W_n(l, l'), \qquad \tilde{f}_n(l, l', 0) := f_n(l, l', 0)(1 - W_n(l, l')). \tag{11}$$

**Assumption 3.** *We assume the the functions $\tilde{f}_n(l, l', 1)$, $\tilde{f}_n(l, l', 0)$ and $\tilde{g}_n(l)$ are uniformly bounded above by $M$ and away from zero by $M^{-1}$ for some constant $M \in (0, \infty)$. We also suppose that either a) there exists a partition $\mathcal{Q}$ of $[0, 1]$ into $\kappa$ parts such that $\tilde{f}_n(l, l', 1)$ and $\tilde{f}_n(l, l', 0)$ are piecewise constant on $\mathcal{Q} \times \mathcal{Q}$, and $\tilde{g}_n(l)$ is piecewise constant on $\mathcal{Q}$; or b) the $\tilde{f}_n(l, l', 1)$ and $\tilde{f}_n(l, l', 0)$ are Hölder($[0, 1]^2$, $\beta$, $L$), and that the $\tilde{g}_n(l)$ are Hölder($[0, 1]$, $\beta$, $L$).*

Assumption 3 will follow as a consequence of Assumptions 1 and 2 for the sampling schemes discussed in Section 3.2, with $\beta$ depending on $\beta_W$ and the hyper-parameters of the sampling scheme.

## 3 Theoretical results

Our theoretical results take the following flavour: we identify the correct population versions (in reference to an infinite graphon on a vertex set $\mathbb{N}$) of $\mathcal{R}_n(\boldsymbol{\omega}_n)$ and $\mathcal{R}_n^{\text{reg}}(\boldsymbol{\omega}_n)$ in the large sample limit $n \to \infty$ to give a regularized population risk, and then use this to give guarantees about any minimizers of $\mathcal{R}_n(\boldsymbol{\omega}_n) + \xi_n \mathcal{R}_n^{\text{reg}}(\boldsymbol{\omega}_n)$ being close (in some sense) to the unique minimizer of the regularized population risk. As network embedding methods are used on very large networks, such a large sample statistical analysis is appropriate.

We introduce the population versions of $\mathcal{R}_n(\boldsymbol{\omega}_n)$ and $\mathcal{R}_n^{\text{reg}}(\boldsymbol{\omega}_n)$ respectively as

$$\mathcal{I}_n[K] := \int_{[0,1]^2} \sum_{x \in \{0,1\}} \tilde{f}_n(l, l', x) \ell(K(l, l'), x) \, dl dl', \qquad \mathcal{I}_n^{\text{reg}}[K] := \int_0^1 K(l, l) \tilde{g}_n(l) \, dl, \quad (12)$$

defined over functions $K : [0, 1]^2 \to \mathbb{R}$. $\mathcal{I}_n[K]$ was first introduced in [21]. The formula given for $\mathcal{I}_n^{\text{reg}}[K]$ holds only for $K$ whose diagonal is well defined; in general, if $K$ admits a decomposition

$$K(l, l') = \sum_{i=1}^{\infty} \mu_i(K) \psi_i(l) \psi_i(l') \quad \text{where} \quad \int_0^1 \psi_i(l) \psi_j(l) \tilde{g}_n(l) \, dl = \begin{cases} 1 & \text{if } i = j, \\ 0 & \text{otherwise,} \end{cases} \quad (13)$$

and $\mu_i(K) \geq 0$ for all $i$ (understood as a limit in $L^2([0, 1]^2)$), then we can extend the definition of the regularizer to be $\mathcal{I}_n^{\text{reg}}[K] := \sum_{i=1}^{\infty} \mu_i(K)$. Consequently, the penalty corresponds to the trace of $K$, when viewed as the kernel of a Hilbert-Schmidt operator. This means that $\mathcal{I}_n^{\text{reg}}[K]$ should be viewed as a nuclear-norm penalty on the kernel $K$, which encourages the $\mu_i(K)$ to be shrunk exactly towards zero, similar to the finite dimensional scenario; also see e.g. Theorem 5.

### 3.1 Guarantees on the learned embedding vectors

We begin with a result which guarantees that $\mathcal{I}_n[K] + \xi_n \mathcal{I}_n^{\text{reg}}[K]$, once restricted to an appropriate domain, is the correct population version of $\mathcal{R}_n(\boldsymbol{\omega}_n) + \xi_n \mathcal{R}_n^{\text{reg}}(\boldsymbol{\omega}_n)$.

**Theorem 1.** *Suppose that Assumptions 2 and 3 hold, and that $\xi_n = O(1)$. Define*

$$\mathcal{Z}_d^{\geq 0}(A) := \left\{ K \, : \, K(l, l') = \langle \eta(l), \eta(l') \rangle, \eta : [0, 1] \to [-A, A]^d \right\} \text{ for } d \in \mathbb{N}, A > 0.$$

*Then we have that*

$$\left| \min_{\boldsymbol{\omega}_n \in ([-A,A]^d)^n} \left\{ \mathcal{R}_n(\boldsymbol{\omega}_n) + \xi_n \mathcal{R}_n^{reg}(\boldsymbol{\omega}_n) \right\} - \min_{K \in \mathcal{Z}_d^{\geq 0}(A)} \left\{ \mathcal{I}_n[K] + \xi_n \mathcal{I}_n^{reg}[K] \right\} \right| = O_p(r_n)$$

*where $r_n = s_n + (d^p/n\rho_n)^{1/2} + t_n$, with $t_n = (\log \kappa/n)^{1/2}$ under Assumption 3a), $t_n = (\log(n)/n^{2\beta/(1+2\beta)})^{1/2}$ under Assumption 3b), $p = 3$ for the cross-entropy loss and $p = 5$ for the squared loss.*

See Appendix C for the proof and a discussion of the rates given; we note that it is necessary that $d \ll n$ in order for $r_n \to 0$. The rates can be improved to give $p = 1$, under additional restrictions on the parameter space (Remark 3). The interpretation of the set $\mathcal{Z}_d^{\geq 0}(A)$ is as follows: if $\omega_u \in [-A, A]^d$ is the embedding of vertex $u$, in the population limit $\eta(\lambda_u) \in [-A, A]^d$ should give the embedding of a vertex with latent feature $\lambda_u$. As (10) is parameterized through terms of the form $\langle \omega_u, \omega_v \rangle$, the population version of (10) should be parameterized through functions $K(\lambda_u, \lambda_v) = \langle \eta(\lambda_u), \eta(\lambda_v) \rangle$.

**Remark 2.** *We note that the assumption that the embedding vectors belong to a hypercube $[-A, A]^d$ is not restrictive; for example, in practice embedding vectors are usually initialized randomly and uniformly over $[-1, 1]^d$. Moreover, our results allow for $A$ to grow logarithmically with $n$, and only change the bound by a poly-log factor. We highlight that if $\xi_n \to \infty$ as $n \to \infty$, then the embedding vectors will shrink towards $0$ as $n \to \infty$ (as seen in Figure 1). This is because the proof of Theorem 1 shows that any minimizer must satisfy $n^{-1} \sum_{i=1}^n \|\omega_i\|_2^2 = O_p(\xi_n^{-1}) = o_p(1)$ in such a regime.*

We now give convergence guarantees for any sequence of embedding vectors minimizing (10).

**Theorem 2.** *Suppose that Assumptions 2 and 3 hold and that $\xi_n = O(1)$. Then for each $n$, there exists a unique minimizer to the optimization problem*

$$K_n^* = \underset{K \in \mathcal{Z}^{\geq 0}}{\arg\min} \left\{ \mathcal{I}_n[K] + \xi_n \mathcal{I}_n^{reg}[K] \right\} \quad \text{where} \quad \mathcal{Z}^{\geq 0} := \text{cl}\Big( \bigcup_{d \geq 1} \mathcal{Z}_d^{\geq 0}(A) \Big)$$

*is free of $A > 0$ (see Proposition 2 for details). Moreover, under some regularity conditions on the $K_n^*$ (see Theorem 7 in Appendix D), there exists $A' < \infty$ free of $n$ and a sequence of embedding dimensions $d = d(n) \ll n$ such that, for any sequence of minimizers*

$$\widehat{\boldsymbol{\omega}}_n \in \underset{\boldsymbol{\omega}_n \in ([-A_1, A_1]^d)^n}{\arg\min} \left\{ \mathcal{R}_n(\boldsymbol{\omega}_n) + \xi_n \mathcal{R}_n^{reg}(\boldsymbol{\omega}_n) \right\} \text{ satisfying } \max_{i,j} |\langle \widehat{\omega}_i, \widehat{\omega}_j \rangle| \leq A_2$$

*with $A_1, A_2 \geq A'$, we have that*

$$\frac{1}{n^2} \sum_{i,j \in [n]} \left( \langle \widehat{\omega}_i, \widehat{\omega}_j \rangle - K_n^*(\lambda_i, \lambda_j) \right)^2 = o_p(1).$$

*Moreover, when Assumption 3a) holds, $K_n^*$ can be computed via a finite dimensional convex program, and is of rank $r \leq \kappa$, in that an expansion of the form (13) holds with $\mu_i(K_n^*) = 0$ for $i > r$.*

The case where $\xi_n = 0$ is proven in [21], which also verifies the convergence on simulated data. A proof is given in Appendix D. Under certain circumstances, this allows us to give guarantees about the distribution of the embedding vectors themselves.

**Theorem 3.** *Suppose that $K_n^*$ is regular in the sense of Theorem 2, with the conclusions of the theorem holding. Moreover suppose that $K_n^*$ is a kernel of rank $r < \infty$, has a decomposition of the form $K_n^*(l, l') = \sum_{i=1}^r \phi_{n,i}(l) \phi_{n,i}(l')$ for some functions $\phi_{n,i} : [0, 1] \to \mathbb{R}$, and the dimension $d$ of the embedding vectors is chosen to be equal to $r$. Writing $\phi_n(l) = (\phi_{n,i}(l))_{i=1}^r$, we have*

$$\min_{Q \in O(r)} \frac{1}{n} \sum_{i=1}^n \left\| \widehat{\omega}_i - Q \phi_n(\lambda_i) \right\|_2^2 = o_p(1). \tag{14}$$

The assumption that $d = r$ in Theorem 3 is a restrictive one, given that embedding dimensions in practice are usually chosen to be one of either 128, 256 or 512. This can be alleviated by instead giving a guarantee for the optimal $r$ dimensional projection of the embedding vectors.

**Theorem 4.** *If instead $d > r$ in Theorem 3, let $\widetilde{G} \in \mathbb{R}^{n \times n}$ denote the best rank $r$ approximation to the matrix $G_{ij} := (\langle \widehat{\omega}_i, \widehat{\omega}_j \rangle)_{ij}$, and write $\widetilde{G} = \widetilde{\Omega} \widetilde{\Omega}^T$ for some $\widetilde{\Omega} \in \mathbb{R}^{n \times r}$. Then $n^{-2} \|\widetilde{G} - G\|_F^2 = o_p(1)$, and writing $\widetilde{\omega}_i$ for the rows of $\widetilde{\Omega}$, the guarantee in (14) holds with $\widetilde{\omega}_i$ replacing the $\widehat{\omega}_i$.*

Informally, this says that the embedding vectors approximately lie on a $r$-dimensional subspace which contains some latent information about the network, depending on the minimizing kernel $K_n^*$. We now highlight that the assumption that $K_n^*$ is of finite rank $r < \infty$ is not restrictive even when $W$ is not a SBM; this is a consequence of the effect of the regularization penalty $\mathcal{I}_n^{\text{reg}}[K]$.

**Theorem 5.** *Let $\ell(y, x)$ be the squared loss, and suppose that $\rho_n = 1$, $\tilde{f}_n(l, l', 1) = \tilde{f}_n(l, l', 0) = c_1$ and $\tilde{g}_n(l) = c_2$ for some constants $c_1, c_2 > 0$ (see e.g. Algorithm 1 in Section 3.2). Then if $W$ is Hölder$([0, 1]^2, \beta, L)$, the minima of $\mathcal{I}_n[K] + \xi_n \mathcal{I}_n^{reg}[K]$ over $\mathcal{Z}^{\geq 0}$ is of finite rank for any $\xi_n > 0$, and is also Hölder continuous of exponent $\beta$.*

We highlight that this result also shows that the regularizer acts to shrink the singular values of a minimizer of $\mathcal{I}_n[K] + \xi_n \mathcal{I}_n^{\text{reg}}[K]$ exactly towards zero. See Appendix E for proofs of Theorems 3, 4 and 5.

## 3.2 Sampling schemes satisfying Assumption 2

We now discuss some examples of frequently used sampling schemes which satisfy Assumption 2. Proofs of the results in this section can be found in Appendix F. We introduce the notation

$$W(\lambda, \cdot) := \int_0^1 W(\lambda, y) \, dy, \quad \mathcal{E}_W(\alpha) := \int_0^1 W(\lambda, \cdot)^\alpha \, d\lambda, \quad \mathcal{E}_W := \mathcal{E}_W(1). \qquad (15)$$

**Algorithm 1** (Uniform vertex sampling). *Given a graph $\mathcal{G}_n$ and number of samples $k$, we select $k$ vertices from $\mathcal{G}_n$ uniformly and without replacement, and then return $S(\mathcal{G}_n)$ as the induced subgraph using these sampled vertices.*

**Lemma 1.** *For Algorithm 1, Assumption 2 holds with $f_n(\lambda_i, \lambda_j, a_{ij}) = k(k-1)$, $\tilde{g}_n(\lambda_i) = k$ and $s_n = 1/n$.*

**Algorithm 2** (Uniform edge sampling [58]). *Given a graph $\mathcal{G}_n$, number of edges to sample $k$, and number of negative samples $l$ per positive sample,*

  i) *We form a sample $S_0(\mathcal{G}_n)$ by sampling $k$ edges from $\mathcal{G}_n$ uniformly and without replacement;*

  ii) *We form a sample set of negative samples $S_{ns}(\mathcal{G}_n)$ by drawing, for each $u \in \mathcal{V}(S_0(\mathcal{G}_n))$, $l$ vertices $v_1, \ldots, v_l$ i.i.d according to the unigram distribution*

  $$\mathrm{Ug}_\alpha(v \,|\, \mathcal{G}_n) \propto \mathbb{P}(v \in \mathcal{V}(S_0(\mathcal{G}_n)) \,|\, \mathcal{G}_n)^\alpha$$

  *and then adjoining $(u, v_i) \to S_{ns}(\mathcal{G}_n)$ if $a_{uv_i} = 0$.*

*We then return $S(\mathcal{G}_n)$ as the union of $S_0(\mathcal{G}_n)$ and $S_{ns}(\mathcal{G}_n)$.*

**Lemma 2.** *For Algorithm 2, Assumption 2 holds with $s_n = (\log(n)/n\rho_n)^{-1/2}$,*

$$f_n(\lambda_i, \lambda_j, 1) = \frac{2k}{\mathcal{E}_W \rho_n}, \ \ f_n(\lambda_i, \lambda_j, 0) = \frac{2kl}{\mathcal{E}_W \mathcal{E}_W(\alpha)} \big\{ W(\lambda_i, \cdot) W(\lambda_j, \cdot)^\alpha + W(\lambda_i, \cdot)^\alpha W(\lambda_j, \cdot) \big\},$$

$$\tilde{g}_n(\lambda_i) = \frac{2kW(\lambda_i, \cdot)}{\mathcal{E}_W} + \frac{2klW(\lambda_i, \cdot)^\alpha}{\mathcal{E}_W \mathcal{E}_W(\alpha)} \cdot \int_0^1 (1 - \rho_n W(\lambda_i, y)) W(y, \cdot) \, dy.$$

**Algorithm 3** (Random walk sampling [25, 48]). *Given a graph $\mathcal{G}_n$, a walk length $k$, number of negative samples $l$ per positively sampled vertex and unigram parameter $\alpha$, we*

  i) *Perform a simple random walk on $\mathcal{G}_n$ of length $k$, beginning from its stationary distribution, to form a path $(\tilde{v}_i)_{i \leq k+1}$, and report $(\tilde{v}_i, \tilde{v}_{i+1})$ for $i \leq k$ as part of $S_0(\mathcal{G}_n)$;*

  ii) *For each vertex $\tilde{v}_i$, we select $l$ vertices $(\eta_j)_{j \leq l}$ independently and identically according to the unigram distribution*

  $$\mathrm{Ug}_\alpha(v \,|\, \mathcal{G}_n) \propto \mathbb{P}(\tilde{v}_i = v \text{ for some } i \leq k \,|\, \mathcal{G}_n)^\alpha$$

  *and then form $S_{ns}(\mathcal{G}_n)$ as the collection of vertex pairs $(\tilde{v}_i, \eta_j)$ which are not an edge in $\mathcal{G}_n$.*

*We then return $S(\mathcal{G}_n)$ as the union of $S_0(\mathcal{G}_n)$ and $S_{ns}(\mathcal{G}_n)$.*

**Lemma 3.** *For Algorithm 3, Assumption 2 holds with $s_n = (\log(n)/n\rho_n)^{1/2}$,*

$$f_n(\lambda_i, \lambda_j, 1) = \frac{2k}{\mathcal{E}_W \rho_n}, \ \ f_n(\lambda_i, \lambda_j, 0) = \frac{l(k+1)}{\mathcal{E}_W \mathcal{E}_W(\alpha)} \big\{ W(\lambda_i, \cdot) W(\lambda_j, \cdot)^\alpha + W(\lambda_i, \cdot)^\alpha W(\lambda_j, \cdot) \big\},$$

$$\tilde{g}_n(\lambda_i) = \frac{kW(\lambda_i, \cdot)}{\mathcal{E}_W} + \frac{(k+1)lW(\lambda_i, \cdot)^\alpha}{\mathcal{E}_W(\alpha)\mathcal{E}_W} \cdot \int_0^1 (1 - \rho_n W(\lambda_i, y)) W(y, \cdot) \, dy.$$

Examining the formula for $\tilde{g}_n(\lambda)$ above, we see that for random walk samplers the shrinkage provided to the learned kernel will be greater for vertices with larger degrees. Indeed, as $K(\lambda, \lambda) = \|\eta(\lambda)\|_2^2$ for $K \in \mathcal{Z}_{\tilde{d}}^{\geq 0}(A)$, the larger $\tilde{g}_n(\lambda)$ is, $\|\eta(\lambda)\|_2^2$ will be forced closer towards zero.

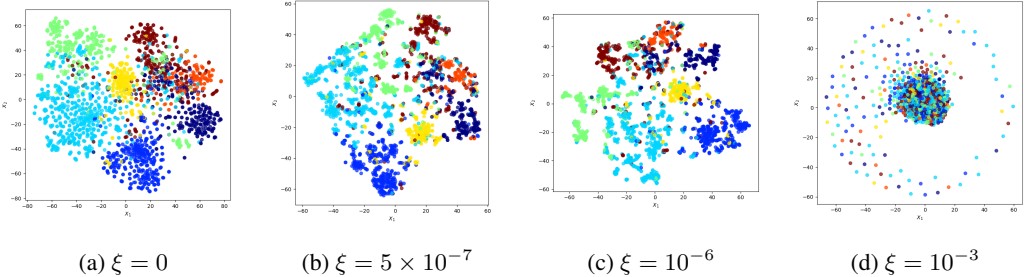

<p align="center">(a) $\xi = 0$     (b) $\xi = 5 \times 10^{-7}$     (c) $\xi = 10^{-6}$     (d) $\xi = 10^{-3}$</p>

Figure 1: TSNE visualizations of Cora network embeddings, learnt using node2vec for different regularization penalties $\xi$, with different colors representing different classes. As $\xi$ increases, the classes begin to cluster together and separate, and then eventually collapse towards the origin.

### 3.2.1 An illustrating example

We now give a brief illustration of our theoretical results under a simple graphon model. To do so, we consider a sparsified SBM with $\kappa$ communities, each equiprobable, and with probabilities $\rho_n p$ and $\rho_n q$ ($p > q$) denoting the within-community and between-community edge probabilities. Writing $A_i = [(i-1)/\kappa, i/\kappa)$ for $i \in [\kappa]$, this can be represented as a graphon model with graphon $W_n = \rho_n W$, where $W(u, v) = p$ if $(u, v) \in \prod_{i=1}^{\kappa} A_i \times A_i$ and $W(u, v) = q$ otherwise.

**Theorem 6.** *Suppose that the graph arises from the model discussed above, we use a cross-entropy loss and the random walk sampling scheme as described in Algorithm 3. Then Theorem 2 holds such that, for any minimizer $\boldsymbol{\omega}_n = (\widehat{\omega}_1, \ldots, \widehat{\omega}_n)$ satisfying (2) of Theorem 2, we have that*

$$\frac{1}{n^2} \sum_{i,j} \left( \langle \widehat{\omega}_i, \widehat{\omega}_j \rangle - K_n(\lambda_i, \lambda_j) \right)^2 = o_p(1) \text{ where } K_n(u, v) = (\tilde{K}_n)_{i,j} \text{ if } (u, v) \in A_i \times A_j,$$

*and $\tilde{K}_n \in \mathbb{R}^{\kappa \times \kappa}$ is defined as the unique positive semi-definite minimizer of the function*

$$-\frac{1}{\kappa^2} \sum_{i,j} \left\{ 2kc_1 \cdot (p\delta_{ij} + q(1 - \delta_{ij})) \log \sigma(\tilde{K}_{ij}) + 2l(k+1) \log \sigma(-\tilde{K}_{ij}) \right\} + \xi c_2 \|\tilde{K}\|_*$$

*where we write $c_1 := ((p + (\kappa - 1)q)/\kappa)^{-1}$ and $c_2 := k + l(k+1)(1 - \rho_n c_1^{-1})$. In particular, for the above example we see that the form of the regularizer is exactly the nuclear norm of $\tilde{K}$, and so as $\xi$ increases, the singular values of the minimizer will be shrunk towards zero.*

## 4 Experiments

We now examine the performance in using regularized node2vec embeddings for link prediction and node classification tasks, and illustrate comparable, when not superior, performance to more complicated encoders for network embeddings. We perform experiments on the Cora, CiteSeer and PubMedDiabetes citation network datasets (see Appendix G for more details), which we use as they are commonly used benchmark datasets - see e.g [26, 28, 34, 64].

### 4.1 Methods used for comparison

For our experiments, we consider 128 dimensional node2vec embeddings learned with either no regularization, or a $\ell_2$ penalty on the embedding vectors with weight $\xi \in \{1, 5\} \times 10^{-\{3,4,5,6,7,8\}}$; and with or without the node features concatenated. We compare these against methods which incorporate the network and covariate structure together in a non-linear fashion. We consider three methods - a two layer GCN [33] with 256 dimensional output embeddings trained in an unsupervised fashion through the node2vec loss, a two layer GraphSAGE architecture [26] with 256 dimensional output embeddings trained through the node2vec loss, and a single layer 256 dimensional GCN trained using Deep Graph Infomax (DGI) [64]. We highlight that the GCN, GraphSAGE and DGI always have access to the nodal features during training. All of our experiments used the Stellargraph [20] implementations for each method. Further experimental details are given in Appendix G.

Table 1: PR AUC scores for link prediction experiments, and macro F1 scores for node classification experiments. "n2v" stands for node2vec without regularization; "rn2v" stands for regularized node2vec with the best score over the specified grid of penalty values; "NF" indicates that node features were concatenated to the learned node embeddings.

| Methods | PR AUC (link prediction) | | | Macro F1 (node classification) | | |
|---------|------|----------|--------|------|----------|--------|
|         | Cora | CiteSeer | PubMed | Cora | CiteSeer | PubMed |
| n2v     | $0.84 \pm 0.01$ | $0.80 \pm 0.02$ | $0.86 \pm 0.01$ | $0.67 \pm 0.04$ | $0.48 \pm 0.03$ | $0.76 \pm 0.00$ |
| rn2v    | $0.90 \pm 0.01$ | $0.88 \pm 0.02$ | $0.91 \pm 0.00$ | $0.73 \pm 0.03$ | $0.55 \pm 0.04$ | $0.77 \pm 0.00$ |
| n2v+NF  | $0.88 \pm 0.01$ | $0.90 \pm 0.01$ | $0.92 \pm 0.00$ | $0.70 \pm 0.03$ | $0.54 \pm 0.03$ | $0.79 \pm 0.01$ |
| rn2v+NF | $0.92 \pm 0.01$ | $0.93 \pm 0.01$ | $0.95 \pm 0.00$ | $0.74 \pm 0.03$ | $0.58 \pm 0.04$ | $0.84 \pm 0.00$ |
| GCN     | $0.90 \pm 0.01$ | $0.87 \pm 0.02$ | $0.94 \pm 0.00$ | $0.67 \pm 0.04$ | $0.48 \pm 0.04$ | $0.80 \pm 0.01$ |
| GSAGE   | $0.90 \pm 0.01$ | $0.90 \pm 0.01$ | $0.88 \pm 0.01$ | $0.74 \pm 0.04$ | $0.56 \pm 0.03$ | $0.79 \pm 0.00$ |
| DGI     | $0.91 \pm 0.01$ | $0.93 \pm 0.01$ | $0.95 \pm 0.00$ | $0.76 \pm 0.03$ | $0.60 \pm 0.02$ | $0.84 \pm 0.00$ |

## 4.2 Link prediction experiments

For the link prediction experiments, we create a training graph by removing 10% of both the edges and non-edges within the network, and use this to learn an embedding of the network. We then form link embeddings by taking the entry-wise product of the corresponding node embeddings, use 10% of the held-out edges to build a logistic classifier for the link categories, and then evaluate the performance on the remaining edges, repeating this process 50 times.

The PR AUC scores are given in Table 1 and visualized in Figure 2. Provided the regularization parameter is chosen optimally, we see an improvement in performance compared to using no regularization, with the jump in performance slightly greater when nodal features are not incorporated into the embedding. The optimally regularized version of node2vec, even without including the node features, is competitive with the GCN and GraphSAGE (being equal or outperforming at least one of them across the three datasets), and that the version with concatenated node features is as good as the GCN trained using DGI. For all the datasets, we observe a sharp decrease in performance after the optimal weight, suggesting that it needs to be chosen carefully to avoid removing the informative network structure. As seen in Figure 1, this occurs as the learned embeddings become randomly distributed at the origin once the regularization weight is too large.

## 4.3 Node classification experiments

To evaluate performance for the node classification task, we learn a network embedding without access to the node labels, and then learn/evaluate a one-versus-rest multinomial node classifier using 5%/95% stratified training/test splits of the node labels. We repeat this over 25 training runs of the embeddings, and a further 25 training/test splits for the node classifiers per embedding. Table 1 and Figure 2 show the average macro F1 scores and their standard deviation for each method and dataset. Similar to the link prediction experiments, we see that the optimally regularized node2vec methods are competitive, if not outperforming, the GCN and GraphSAGE trained through the node2vec loss, and is outperformed by the GCN learned using DGI by no more than two percentage points. In these experiments, the standard deviations correspond partially to the randomness induced by the training/test splits of the node labels, and suggest that the regularized version of node2vec is no less robust to particular choices of training/test splits than the other methods.

Interestingly, we note that the optimal performance on PubMed is given by the regularized node2vec with node features. However, as the highest level of performance is performed when the regularization weight is so large that the learnt embeddings are uninformative (as illustrated by the massive decrease in performance of the regularized method without node features), this suggests that the nodes can be classified using only the covariate information, and that the network features are not needed. As the dataset only has three distinct classes corresponding to the academic field, and the node features are TF-IDF embeddings of the academic papers, this is not too surprising.

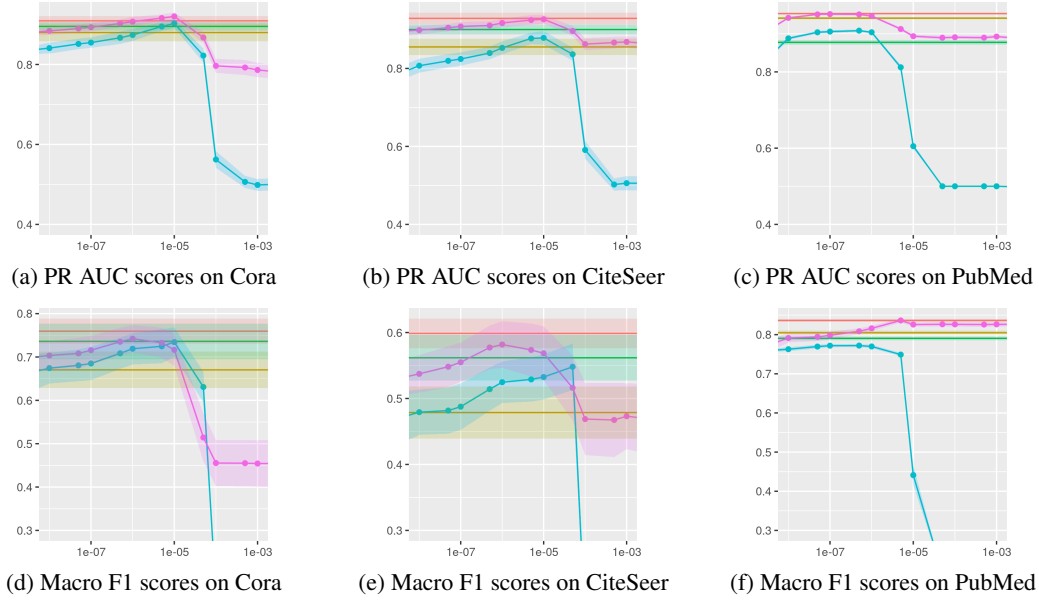

Figure 2: Plots of average PR AUC scores for link prediction and macro F1 scores for node classification (bands representing plus/minus one standard deviation), against the regularization penalty used for node2vec with (purple) and without (blue) concatenated node features. The results from GCN, GraphSAGE and DGI are given as horizontal bars in yellow, green and red respectively.

## 5 Conclusion

In this paper we have theoretically described the effects of performing $\ell_2$ regularization on network embedding vectors learned by schemes such as node2vec, describing the asymptotic distribution of the embedding vectors learned through such schemes, and showed that the regularization helps to reduce the effective dimensionality of the learned embeddings by penalizing the singular values of the limiting distribution of the embeddings towards zero. We do so in the non-convex regime $d \ll n$, reflecting how embedding dimensions are chosen in the real world. We moreover highlight empirically that using $\ell_2$ regularization with node2vec leads to competitive performance on downstream tasks, when compared to embeddings produced from more recent encoding and training architectures.

We end with a brief discussion of some limitations of our works, and directions for future work. Generally, graphons are not realistic models for graphs; we suggest that our work could be extended to frameworks such as graphexes [10, 11, 62] which can produce more realistic degree distributions for networks, but have enough underlying latent exchangeability for our arguments to go through. We ignore aspects of optimization, i.e whether the minima of (10) are obtained in practice, which we believe would be an interesting area of future research. As for our experimental results, we note that methods such as GraphSAGE are better than node2vec in that they provide embeddings for data unobserved during training, and also scale better to larger networks. Consequently, we believe our experiments should be used primarily as motivation to investigate better methods for incorporating nodal covariates into network embedding models, and how to regularize embeddings produced by encoder methods such as GCNs or GraphSAGE.

## Acknowledgments and Disclosure of Funding

We thank Morgane Austern for helpful discussions, along with several anonymous reviewers for their comments on the current version of the paper, along with a prior version. We also acknowledge computing resources from Columbia University's Shared Research Computing Facility project, which is supported by NIH Research Facility Improvement Grant 1G20RR030893-01, and associated funds from the New York State Empire State Development, Division of Science Technology and Innovation (NYSTAR) Contract C090171, both awarded April 15, 2010.

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
