# OpenReview forum: "Asymptotics of $\ell_2$ Regularized Network Embeddings"
_NeurIPS.cc/2022/Conference — NeurIPS 2022 Accept_

### Official Review · Reviewer_ocw3 · 2022-07-11

**Rating:** 7
**Confidence:** 2
**Soundness:** 4 excellent
**Presentation:** 3 good
**Contribution:** 4 excellent

**Summary:**

The authors start with background on skip-gram node embedding methods like DeepWalk and node2vec, then review some prior theoretical work on guarantees for embedding methods. They then establish some fairly strong assumptions for the following analysis. The bulk of the paper concerns establishing new guarantees for embedding methods, improving on prior guarantees by additionally handling $\ell_2$ regularization, which is commonly used and known to increase performance; other theory work includes partially justifying the assumptions by showing that common sampling schemes for embedding methods satisfy some of them. Finally, some experiments on real-world datasets establish that 1) there is a sweet spot of regularization strength for best performance of node2vec on downstream tasks; 2) performance is generally improved by concatenating node features to embeddings; and 3) performance is often competitive with deep methods.

**Questions:**

I am not familiar with the "Asymptotics of Network Embeddings Learned via Subsampling" paper, which is cited here as studying a similar non-convex setting with $d \ll n$, but without regularization. The authors note some similarities in the analyses, but I cannot judge the extent to which the theory work here is original. Some discussion of how the regularization requires new proof steps/techniques may be of interest.

The setting here seems to concern methods where a single embedding vector is produced per node, and the probability of an edge between two nodes increases monotonically with the dot product of their embeddings. Limitations of this setting have been discussed previously ("The impossibility of low-rank representations for triangle-rich complex networks," "Node Embeddings and Exact Low-Rank Representations of Complex Networks," among others). DeepWalk and node2vec themselves produce two embeddings per node ("word" and "context" embeddings, per word2vec's terminology), though they may discard one or aggregate them in the end. Do the authors have any comment on how this affects the analysis, and whether extensions to the two-embedding / non-PSD setting are possible?

Minor comments:
+ Line 31: uses $N$ for negative sampling function, while Equation 1 uses $\mathcal{N}$
+ Line 40: same as above
+ "e.g" vs "e.g." (not necessarily a mistake, but NeurIPS seems to use the latter, e.g., in the checklist instructions)
+ "node covariates" and "node features" are both used interchangeably throughout the paper; I rarely see the former, so sticking to the latter may improve clarity
+ a graphical legend for Figure 2 would improve readability a bit, though the caption and trends in the plot are clear enough as is
+ Is $n$ used both for the number of nodes and indexing the sequence of graphs/graphons? If so, it is likely worth changing one for clarity (perhaps $t$?).

**Limitations:**

The others note some limitations of their work in the conclusion, including certain strong assumptions required for the proof, which seems sufficient. They decline to discuss potential negative societal impacts given the theoretical nature of the work, saying that it would be excessively speculative. Though some generic discussion of potential impacts seems possible based on the NeurIPS guidelines, their viewpoint is understandable.

**Strengths And Weaknesses:**

Strengths
+ The writing is well-structured and grammatical, if a bit dense, though that may be inevitable given the technicality of this work.
+ The work advances theory for node embeddings learned with regularization, for which better theoretical understanding is of great interest to graph machine learning.
+ Though I did not go through the extensive appendix or even understand all parts of the main paper, the parts that I did understand generally seem sound.

Weaknesses
+ As the authors note, the assumptions required for the proofs are quite strong and preclude some common characteristics of real-world graphs.

Notes (Neutral)
+ The paper is quite technical and assumes a lot of prior knowledge. If accepted, it would increase the reach of the paper if the extra page were used to add some background/intuition on exchangeability, graphons, etc. and how they are used in the assumptions/proofs, perhaps using the specific case of SBMs (which does seem to arise in the appendix) for a concrete example.
+ It is difficult for me to judge the originality of the work, as I discuss in the Questions section.
+ The empirical section does not seem particularly original or specific to the main work of this paper; it is generally the case that there will be some sweet spot of regularization strength for downstream performance. It would be more interesting if an experiment somehow validated or demonstrated the theoretical findings.

---

> ### Author Response · Authors · 2022-07-29
> **Reply to reviewer**
>
> We would like to begin by thanking you for your time and efforts in reviewing our paper. To respond to your questions:
>
> > I am not familiar with the "Asymptotics of Network Embeddings Learned via Subsampling" paper, which is cited here as studying a similar non-convex setting with , but without regularization. The authors note some similarities in the analyses, but I cannot judge the extent to which the theory work here is original. Some discussion of how the regularization requires new proof steps/techniques may be of interest.
>
> We appreciate the suggestion to clarify the distinction between our work and the paper mentioned. The theory work in our paper requires new proof techniques in order to argue for the existence of a (unique) minimizer of the regularized population risk. As the regularization acts as a trace/nuclear norm penalty on the kernel $K: [0, 1]^2 \to \mathbb{R}$, it becomes necessary to argue that $\mathcal{I}_n[K] + \xi_n \mathcal{I}_n^{\text{reg}}[K]$ is lower semi-continuous and has compact level sets in a suitable topology defined on the space of kernels which are of finite trace; different arguments are needed in the non-regularized setting as the domain is a different space. As a result, Propositions 2 and 3 are novel. The proof arguments for Theorems 1 and 2 are similar, with some minor technical modifications; the most major modification is in Lemma 8, which arises from considering the KKT conditions for $\mathcal{I}_n^{\text{reg}}[K]$. Lemma 11 and the derivations of the functions $g(\lambda)$ for the sampling schemes given in Section E are also a novel addition.
>
> From a results perspective, our analysis also gives stronger guarantees than the aforementioned paper, as here we give guarantees of the form
> $$ \frac{1}{n^2} \sum_{i, j \in [n] } \big|  \langle \widehat{\omega}_i, \widehat{\omega}_j \rangle - K(\lambda_i, \lambda_j) \big|^p = o_p(1)$$
> for $p = 2$, whereas the former paper only gives guarantees for $p = 1$. This arises from a more careful analysis in Lemma 8 than the analogue in the other paper. This then allows us to give the types of guarantees stated in Theorems 3-5 of our paper, which are novel and would not be possible otherwise.
>
> > The setting here seems to concern methods where a single embedding vector is produced per node, and the probability of an edge between two nodes increases monotonically with the dot product of their embeddings. Limitations of this setting have been discussed previously [...]. DeepWalk and node2vec themselves produce two embeddings per node ("word" and "context" embeddings, per word2vec's terminology), though they may discard one or aggregate them in the end. Do the authors have any comment on how this affects the analysis, and whether extensions to the two-embedding / non-PSD setting are possible?
>
> This is an interesting question! To introduce some notation, consider the two-embedding setting where one has embeddings $\omega_u$ and $\tilde{\omega}_u$ for each node $u$. Equation (6) - for example - would then change by replacing the terms $\langle \omega_i, \omega_j \rangle$ with terms $\langle \omega_i, \tilde{\omega}_j \rangle$, and the $\ell_2$ penalty would become
>
> $$ \sum_{i=1}^n \big( || \omega_i ||_2^2 + || \tilde{\omega}_i ||_2^2 \big) = || \Omega ||_F^2 + || \widetilde{\Omega} ||_F^2$$
> where we write $\Omega, \widetilde{\Omega} \in \mathbb{R}^{n \times d}$ for the matrices whose rows are the $\omega_i$ and $\tilde{\omega}_i$ respectively. We note that for a rank $d$ matrix $G \in \mathbb{R}^{n \times n}$, one has
>
> $$ || G ||_{*} = \min \frac{1}{2} \big( || U ||_F^2 + || V ||_F^2 \big)  \text{ s.t } G = UV^T, U, V \in \mathbb{R}^{n \times d}$$
>
> (a similar result holds in the infinite dimensional setting also), and so a similar phenomenon should be observed as in the PSD/single embedding case we discuss. We expect that the arguments in Theorems 1 and 2 could be adapted to this setting with some modifications (we can expand upon this if desired). Theorem 3 (i.e guarantees for the embedding vectors themselves, and not their inner products) could also be generalized, for example by using Lemma 5.14 of [56].
>
> We would like to also thank the reviewer for the suggestion to use the additional page (if the paper is accepted) to add some information about graphon models and to use SBMs as concrete examples - if accepted, this is something which we will use the additional space for. We also appreciate the comments with regards to e.g. typos, terminology and legends, which will be incorporated into the paper. For the last point, $n$ is used for both the number of nodes and the indexing of the sequence of the graphs. As the $n$-th graph in the sequence is always assumed to have $n$ nodes (there is no randomness in the number of nodes in a particular realization of the network), the same variable is used for both.
>
> If any more questions arise during the discussion period, we will be happy to address them. Thanks again!

---

> > ### Comment · Reviewer_ocw3 · 2022-08-08
> > **Response to Authors' Comment**
> >
> > Thank you for the thorough response. I would suggest adding some discussion of this comparison to this prior paper, but I think the extra space would be better spent on more background info and a concrete example as you wrote. I am keeping my score to support the paper.

---

### Official Review · Reviewer_Bk2Z · 2022-07-11

**Rating:** 6
**Confidence:** 2
**Soundness:** 3 good
**Presentation:** 2 fair
**Contribution:** 3 good

**Summary:**

This work studies the effects of adding a l2-penalty of the embedding vectors to the training loss of network embedding methods. Mainly, the work gives guarantees on the asymptotic distribution of learned embedding vectors under certain assumptions. Particularly, when the embedding vectors are sampled i.i.d from some latent distribution, the distribution of adjacency matrices arise from a probabilistic model characterized by a graphon. Riding on this assumption, the authors work with the population versions of the empirical risk function and give guarantees for a unique minimizer for the regularized population risk in the large sample limit.

**Questions:**

It is not clear why an exchangeable graph model and consequently an underlying graphon process for graph generation is a valid starting point. There is not enough motivation in the paper behind this critical assumption.

Further analysis on the role of the sparsifying sequence rho needs to be provided. As the graphon implicitly gives rise to dense graphs, different choices of rho might be influential to model different types of graphs. An analysis relevant to the experimental datasets chosen is lacking.

Theorems 1-5 are very interesting and concisely derived. They show that working with population versions is a promising line of enquiry and even simple prior distributions on the embeddings gives rise to very interesting asymptotic dynamics.

Motivation behind assumption 2 is opaque. It is not perfectly clear to me why this assumption is important and how it helps in arriving at results seen in theorem 1-5.

More details about the relationship between the assumptions in the paper and the network datasets chosen for experimental validation are warranted. These details are currently missing in the experimental results section.

Notes:

Eq 1: \langle is not defined.

31: N and \mathcal{N} are mixed up. This occurs many times. Same with P and \mathcal{P}.

"Correct population version": This is not a precise statement; define "population version".

133: Assumption not explained.

168: "a discussion of the rates give".

Eq 4: a_uv is not defined.

**Strengths And Weaknesses:**

The paper has interesting theoretical results and insightful application of graph theoretical concepts to asymptotic analysis of network embeddings. Theorems have very interesting results and are concisely derived. The paper lacks a clear explanation for the motivation behind the assumptions and impact of the assumptions on the experimental datasets of choice and real world datasets in general. Overall, this work is an interesting line of enquiry and might spur additional investigations along similar lines.

---

> ### Author Response · Authors · 2022-08-02
> **Reply to reviewer**
>
> We would like to begin by thanking you for your time and efforts in reviewing our paper. We also appreciate the comments about some of the assumptions needing further clarifications; we provide some expanded explanations and justifications below (after quoting the relevant comment). If the paper gets accepted, we will use some of the extra space to include these descriptions.
>
> > It is not clear why an exchangeable graph model and consequently an underlying graphon process for graph generation is a valid starting point. There is not enough motivation in the paper behind this critical assumption.
>
> In the introduction, we argue that embedding methods such as node2vec are implicitly fitting a (dense) exchangeable graph model to the data. As a first step to analyzing such embedding methods, we therefore take the statistical perspective of assuming the data are generated from the same family of models, and we include the sparsifying sequence to account for the fact that graphs observed in the real world are rarely dense (even though a dense graph model is being fit implicitly). This is mentioned in the introduction, but this being the reasoning for the assumption of the generative model could have been made more explicit. Another justification is that graphon models are a standard tool used for the analysis of graph ML procedures; for instance spectral clustering using graph Laplacians, or some recent-ish work on over-smoothing in GNNs (https://arxiv.org/abs/1905.10947). The smoothness assumptions are common when making non-parametric assumptions on a model class. We can add words to this affect before/after Assumption 1 to address this point.
>
> > Further analysis on the role of the sparsifying sequence rho needs to be provided. As the graphon implicitly gives rise to dense graphs, different choices of rho might be influential to model different types of graphs. An analysis relevant to the experimental datasets chosen is lacking.
>
> We thank you for the suggestion to explain more the role of the sparsifying
> sequence $\rho_n$. With regards to the conditions this implies on the generated graph, it is known that requiring $\rho_n \gg \log(n)/n$ leads to graphs with a "regular" degree structure (that is, the degrees are all roughly of the same order), which is a limitation which we can add into Remark 1.
>
> > Motivation behind assumption 2 is opaque. It is not perfectly clear to me why this assumption is important and how it helps in arriving at results seen in theorem 1-5.
>
> We thank you for the suggestion to improve the motivation for Assumption 2 in the paper. This assumption allows us to argue that it suffices to consider the minimizers of the function
>
> $$ \frac{1}{n^2} \sum_{i \neq j } f_n(\lambda_i, \lambda_j, a_{ij}) \ell( \langle \omega_i, \omega_j \rangle, a_{ij}) + \frac{1}{n} \sum_{i \in [n] } \tilde{g}_n(\lambda_i ) || \omega_i \|_2^2 $$
>
> instead, by replacing the sampling probabilities with $n^{-2} f_n(\lambda_i, \lambda_j, a_{ij})$ and $n^{-1} \tilde{g}_n(\lambda_i)$ respectively. This function is now amenable to analysis in the limit as the number of vertices $n \to \infty$, as we can explicitly use the randomness and exchangeability in the generative model to analyze the above function.
>
> > More details about the relationship between the assumptions in the paper and the network datasets chosen for experimental validation are warranted. These details are currently missing in the experimental results section.
>
> We appreciate the suggestion to motivate the choice of network datasets. The datasets were chosen as they are commonly used benchmark datasets when comparing different graph machine learning methods, and were used by the papers on GCNs, GraphSAGE and DGI which we used as a point of comparison. In terms of trying to relate the assumptions we make to the network datasets chosen, as we only observe a single fixed realization of a graph, it is hard to argue to whether one particular observed graph could plausibly arise from a graphon model, or whether a graph "mostly" arises from one (i.e, only a few nodes are outliers, and the rest of the graph could plausibly be generated from such a model), or not at all. The same is true looking at the sparsity level of the graph, and trying to compare this to a function $\rho_n$ depending on the number of vertices $n$.
>
> We would also like to thank you for highlighting various typos and terms which need defining; we will fix these and make these additions in the paper. We agree that the meaning of "correct population version" needs expanding upon; here the "population" is in reference to an infinite graph whose vertex set is $\mathbb{N}$.
>
> If any more questions arise during the discussion, we'll be happy to answer them, and any comments about whether the above explanations suffice would be greatly appreciated. Thanks again for your time!

---

### Official Review · Reviewer_xqKB · 2022-07-12

**Rating:** 5
**Confidence:** 1
**Soundness:** 3 good
**Presentation:** 2 fair
**Contribution:** 2 fair

**Summary:**

This paper studies the effects of adding an $\ell_2$ regularizer to classic unsupervised network embedding objectives. This paper analyzes the asymptotic distribution of the embedding vectors learned through such schemes, and showed that the regularization helps to reduce the effective dimensionality of the learned embeddings. This method is simple and effective, and achieves good performance on the citation datasets. In summary, the theoretical analysis of the paper is well presented. The motivations and the authors' insights to this model is well explained. But the experimental part is weak.  It would be nice to have more experiments on versatile and larger datasets.

**Questions:**

$\ell_2$ regularization  is a common operation and the overall enhancement regarding to accuracy over baselines seems very marginal, which makes the whole novelty a little weaker.  It is better for the authors to provide more experiment to demonstrate the advantages of this method, such as extend $\ell_2$ regularization to other network embedding.

**Limitations:**

I don't see any potential negative societal impact.

**Strengths And Weaknesses:**

**Strengths:**

1. Provide some theoretical insights on a simple and commonly used approach.
2. Reasonable motivation.


**Weaknesses**
1. The theoretical results might be useful, but the asymptotical analyses build on many assumptions, which makes the theory less practical. Moreover, the paper is dense, and it is easy to get lost in the very technical definitions and derivations. The paper advocates $\ell_2$ regularization is good, which is well-known empirically, but the theoretical "take away message" is not quite clear based on the presentation.
2. In the experimental part, this article only tests the performance on Cora, Citeseer and Pubmed. These datasets are old and small-scale. I suggest that the author test on large datasets such as Flickr, Reddit and Arxiv. On the other hand, it would be more convincing to use more unsupervised models such as VGAE，MVGRL as baseline models.

---

> ### Author Response · Authors · 2022-08-02
> **Reply to reviewer**
>
> We would like to begin by thanking you for your time and efforts in reviewing our paper. To respond to some parts in the Questions section, and also some comments in the weaknesses section:
>
> > It is better for the authors to provide more experiment to demonstrate the advantages of this method, such as extend $\ell_2$ regularization to other network embedding.
>
> Although we didn't report this in the paper, in some preliminary experiments we tried penalizing the $2$-norm of the outputs of encoders such as GCNs or the GraphSAGE encoder (which corresponds to a given embedding of a particular vertex), and only observed a sharp decrease in performance when doing so. We agree that being able to perform effective dimension reduction for more complicated encoders would be desirable - unfortunately, we aren't aware of a way of doing so currently.
>
> > In the experimental part, this article only tests the performance on Cora, Citeseer and Pubmed. These datasets are old and small-scale. I suggest that the author test on large datasets such as Flickr, Reddit and Arxiv. On the other hand, it would be more convincing to use more unsupervised models such as VGAE, MVGRL as baseline models.
>
> We were also not aware of the MVGRL paper, and so we thank you for making us aware of it. Although we did not report other metrics in the paper (the trends observed were essentially identical), for our experiments we also computed ROC AUC scores for the link prediction experiments, and we obtain similar values (i.e within rounding and the error bars we report) to those reported in the VGAE paper for the Cora, Citeseer and Pubmed datasets. As for the MVGRL method, this appears to have a marginal-to-okay improvement over DGI (and therefore presumably on regularized node2vec) on the Cora, Citeseer and Pubmed datasets, at the tradeoff of using additional information (in the inclusion of the diffusion matrix). We used the Cora, CIteseer and Pubmed datasets as they seem to be still standard (as in e.g the MVGRL paper), but we agree that comparisons on larger datasets would be interesting. We are afraid that we may not have the time to do further experiments before the rebuttal period ends, although if we do, we will report the results.
>
> If any more questions or comments arise, we would be happy to address them. Thanks again for your time!

---

> > ### Comment · Reviewer_xqKB · 2022-08-09
> > **Reply to Rebuttal**
> >
> > Thanks for the response.
> >
> > After reading comments from other reviewers and the authors' response, I understood the material slightly better now and I think there are some useful theoretical insights. Thus, I raise my score. However, I still have some concerns on the empirical studies, which seems weak.

---

### Official Review · Reviewer_XosH · 2022-07-12

**Rating:** 7
**Confidence:** 3
**Soundness:** 4 excellent
**Presentation:** 3 good
**Contribution:** 3 good

**Summary:**

This paper proposes to study from both a practical and theoretical perspective, node embeddings methods whose embeddings are regularized with an $\ell_2$ loss. In particular, the authors propose considering here node embedding methods that, given a graph, embed the nodes in Euclidian space, typically by minimizing a loss over small batches/ subgraphs (eg Node2vec, deepwalk).  The authors show that this corresponds to trying to learn the underlying graphon and latent representations, ie, the function W  and latent representations  $\lambda_u$ such that:
$$  \lambda_u \sim \text{Unif}([0,1]), \quad a_{uv} | \lambda_u, \lambda_v \sim \text{Bernouilli}(W(\lambda_u, \lambda_v))$$
Since this only yields dense graph (Aldous Hoover theorem), this requires to be sparsified, and consider $\rho_n W$ where $\rho_n \to 0$.
The authors note that if we assume the embeddings here (called $\omega_i$)  to be Gaussian, then this is equivalent to adding a Gaussian penalty in the reconstruction of the objective function, and the embedding reconstruction can be interpreted as matching a graphon kernel:
$$ \frac{1}{n^2}  \sum_{ij} (<\omega_i, \omega_j> - K(\lambda_i, \lambda_j))^2 = o_p(1)$$
where $K :[0,1]^2 \to [0,1]$.

The authors then show that the minimum of the population estimates for their risk function should converge to the same as for the risk function;  The paper then details the procedure that the authors suggest to estimate the embeddings: in particular, they suggest a couple of sampling schemes that are aligned with the assumptions that the authors made in the derivations of their theorems.
On the practical/experiments side, the authors show that concatenating their purely structural embedding with node features yields better performance than GNNs.

**Questions:**

NA

**Limitations:**

 These approaches are purely structural: the node features are considered independently of the graph structure. Graphons are further not suitable to the modeling of a number of real-life graphs, including power laws.

**Strengths And Weaknesses:**

Strengths:
- this is a solid paper, that attempts to shed light into the mechanisms underpinning node embeddings. The authors' method is interesting, as it draws ties behind (what we could perhaps qualified as "heuristics-based") approaches in ML and the extensive non-parametric literature on graphon estimation. Tying the two together, and including an $\ell_2$ regularisation as a prior on the embedding is a nice way of drawing the two pieces of literature together.
The paper makes a nice theoretical analysis of structural embeddings --- which can then be added to node features for classification --- and shows good performance in their experiments.


Limitations:
- These approaches are purely structural: the node features are considered independently of the graph structure.



Typos:
-l2: "begin" instead of "begins"

---

> ### Author Response · Authors · 2022-07-29
> **Reply to reviewer**
>
> We would like to thank you for your time and efforts in reviewing our paper, and for highlighting some typos in the paper. If any questions arise between now and the end of the discussion period, we will be happy to answer them.

---

### Comment · Area_Chair_fv3k · 2022-08-05
**Further Questions For Authors**

Hi,

Thanks for the interesting submission and for responding informatively to the reviews.

I had a high level question about the overall theoretical message of the paper. The main claim seems to be that more regularization reduces the effective dimensionality of the learned embeddings. But given that embedding vectors are typically already picked in low dimensions, and also given that we can pick the dimensionality to optimize performance, why is this useful?

Relatedly, why is the graphon based model and analysis of the population limit needed to make this realization? Given work showing that DeepWalk, node2Vec, etc. are essentially performing low-rank matrix factorizations on different graph-derived matrices ("Network embedding as matrix factorization: Unifying deepwalk, line, pte, and node2vec. ") it should follow that placing l_2 penalties on the factors U, V should tend to decrease the nuclear norm of UV^T (since ||UV^T|| ~= min_{U,V} ||U||_F^2 + ||V||_F^2.)

It would be helpful if the authors could address these questions and clarify if I am oversimplifying the main theoretical message.

Thanks!

---

> ### Author Response · Authors · 2022-08-06
> **Reply to further questions**
>
> Thanks for taking the time to look at our submission, and the additional questions! To respond to both:
>
> > I had a high level question about the overall theoretical message of the paper. The main claim seems to be that more regularization reduces the effective dimensionality of the learned embeddings. But given that embedding vectors are typically already picked in low dimensions, and also given that we can pick the dimensionality to optimize performance, why is this useful?
>
> To set some background, in practice the embedding dimensions are chosen in practice to usually be some power of 2 (256 appears to be the most common, with 128 and 512 also seen), and generally aren't varied as hyperparameters of the model (with one or two being chosen at best). This is in contrast to other hyperparameters - in the case of node2vec, these would be parameters like the walk length, the number of negative samples, or the breadth/depth search parameters $p$ and $q$ sometimes used. This is similar as with output representations used in NLP - for example, when fitting models like BART, the output dimension is chosen to be "large enough", rather than tuned optimally. In 'Node Embeddings and Exact Low-Rank Representations of Complex Networks' (https://arxiv.org/abs/2006.05592) it is argued that commonly used network datasets have representations in even lower dimensions - for example, in dimensions $\leq 48$ for the citation network datasets we used for the experiments in our paper - than the embedding dimensions chosen by practitioners.
>
> So far, nothing we've said argues against the idea of just choosing the embedding dimensions as hyperparameters. An advantage to using a regularization weight which shrinks the effective dimensionality, is that it can be tuned in a more computationally efficient manner by using a 'warm-start' procedure (similar to e.g implementations of the LASSO). To be concrete, one can learn the embeddings assuming a given regularization weight, and then use these as initial points for the optimization of a larger regularization weight; this will give computational savings over needing to re-select an embedding dimension each time, as the minima of the different optimization procedures will be close to each other when the regularization weight is close. This will work best when the initial embedding dimension is chosen to overshoot the "actual dimension" of the given network, which is what generally occurs in practice. In contrast, we're not aware of a warm-start procedure when reducing the embedding dimension chosen to fit an embedding model; potentially one could try sketching the matrix of embeddings, but this still has some added cost associated with it.
>
> > Relatedly, why is the graphon based model and analysis of the population limit needed to make this realization? Given work showing that DeepWalk, node2Vec, etc. are essentially performing low-rank matrix factorizations on different graph-derived matrices ("Network embedding as matrix factorization: Unifying deepwalk, line, pte, and node2vec. ") it should follow that placing l_2 penalties on the factors U, V should tend to decrease the nuclear norm of UV^T (since ||UV^T|| ~= min_{U,V} ||U||_F^2 + ||V||_F^2.)
>
> Our paper handles how we can try and preserve this intuition in the presence of using SGD schemes for learning embeddings of networks. Note that for methods such as node2vec, the probability that a vertex appears in a given subsample for a particular gradient estimate is not equal across all vertices - as node2vec performs random walks on a graph to select vertices, vertices of higher degree will be sampled more frequently. In particular, this means that when considering the empirical risk being minimized by these methods, the penalty for the embedding matrices won't be of the form $||U ||_F^2 + || V ||_F^2$, as the embeddings for vertices of higher degree will contribute more frequently than those of lower degree. That said, we would still like to be able to argue that some effective dimension reduction occurs by consideration of a nuclear-norm style penalty; we provide a way of being able to do so. (One of the reviewers mentioned it would be useful to contextualize parts of the paper by using a simple stochastic block model - I think this would be one good place to do so, using some of the extra space if the paper is accepted.)
>
> More generally, the "Network Embedding as Matrix Factorization" paper is good at a level of giving intuition, but makes some assumptions which aren't realistic (such as taking the embedding dimension $d = n$, or taking the walk length $L \to \infty$ when examining node2vec). We instead work under an assumption where $d \ll n$ and $n$ is large, and use the graphon assumption and population analysis to allow us to be rigorous when doing so.
>
> Please let us know if these don't fully address your questions, or if any others come up.

---

> > ### Comment · Area_Chair_fv3k · 2022-08-07
> > **Response**
> >
> > Thanks -- this is helpful. I think it would be helpful to include some of this explanation in the intro of the paper to give some more context to the contribution.

---

### Author Response · Authors · 2022-08-09
**Revision details**

Thanks again to everyone for their comments and suggestions - they have been very useful! We have submitted a revision fixing the typos mentioned by the reviewers, and also addressing the explanations where the reviewers/area chairs have commented that they are either lacking, or should be added to help better contextualize the paper. As we are unsure about whether we can use the additional page at this stage, we have not yet incorporated a longer discussion as to graphon models and to give some examples contextualizing the theoretical results of the paper, but if the paper is accepted, we will use the extra space for this, and expanding other explanations as space permits.

---

### Meta-Review · Area_Chair_fv3k · 2022-08-24

**Recommendation:** Accept
**Confidence:** Certain

**Metareview:**

Overall, reviews on this paper are universally positive. The paper presents an interesting theoretical analysis of l_2 regularized node embeddings via an asymptotic graphon model based approach. The reviewers find this approach interesting, and hopefully the techniques will be valuable in further theoretical progress on understanding node embedding algorithms.

The paper does have some weaknesses, including a somewhat lacking empirical validation and experiments that don't clearly confirm or complement the theoretical insights. The paper could also do a better job contextualizing the main theoretical insight, which seems to be that regularization can be used to control effect dimensionality of the node embeddings. As discussed, connecting this high level conclusion to prior work on analysis of node embeddings as matrix factorization would be valuable.

The reviewers gave helpful feedback on many aspects of the paper and the authors have agreed to include much of this feedback in a revised draft. I hope that the authors will do so, as it will strengthen the presentation significantly.

**Award:**

No

---

### Decision · Program_Chairs · 2022-09-14

Accept